# Improvement of the Water Quality in Rainbow Trout Farming by Means of the Feeding Type and Management over 10 Years (2009–2019)

**DOI:** 10.3390/ani10091541

**Published:** 2020-09-01

**Authors:** Elisa Fiordelmondo, Gian Enrico Magi, Francesca Mariotti, Rigers Bakiu, Alessandra Roncarati

**Affiliations:** 1School of Bioscience and Veterinary Medicine, University of Camerino, Circonvallazione 93-95, 62034 Matelica, Italy; gianenrico.magi@unicam.it (G.E.M.); francesca.mariotti@unicam.it (F.M.); alessandra.roncarati@unicam.it (A.R.); 2Department of Aquaculture, University of Tirana, Place Mother Tereza 183, 695581 Tirana, Albania; bakiurigers@gmail.com

**Keywords:** rainbow trout farming, water quality, feed extrusion technology, total suspended solids

## Abstract

**Simple Summary:**

The European Union Water Framework Directive has set the objective to develop a good status for water bodies and showed the importance of promoting the sustainable use of water resources in inland anthropic activities, such as freshwater aquaculture, performed in strict accordance with natural waters. In aquafeed, two elements have mostly contributed to improving aspects of change in fish feeding among rainbow trout—the use of the feed extrusion technique and the adoption of restrictive environmental rules. In this context, the main physico-chemical parameters of water quality were investigated in 2019 and the values obtained were then compared to the parameters obtained 10 years before (2009) in order to show if there were differences in the water quality of an outlet farm. Considering this, the present study aimed to evaluate the suitability of changes adopted in rearing and feeding techniques to improve growth performance, sustainability and the water quality environment. This study was made possible by considering data sampled on a yearly basis in the outlet and compared with the inlet water.

**Abstract:**

Background: In Europe, rainbow trout is one of the main fresh water fish farmed in a constantly developing environment that requires innovative studies to improve farm management, fish welfare and environmental sustainability. The aim of this paper is to investigate the trend of water quality parameters over 10 years, after a feeding strategy change from pellet to extruded feed. Methods: The study was conducted on a farm in central Italy, based on parallel raceways. The cycle started from young rainbow trout (90 ± 2 g) that were grown until they reached market size. A water sample of 500 cm^3^ was collected monthly from 2009 to 2019 from the lagoon basin in order to investigate the trends of the total suspended solids (TSS), biochemical oxygen demand (BOD_5_), chemical oxygen demand (COD), nitrites (NO_2_-N), nitrates (NO_3_-N), total ammonia nitrogen (TAN), total phosphorus (TP) and pH. Results: All of the studied parameters (TSS, BOD_5_, COD, NO_2_-N, NO_3_-N, TAN and TP) showed a significant improvement from 2009 to 2019. The pH parameter did not display notable variation during the studied period. The feed conversion ratio (FCR) was also investigated and exhibited a significant improvement from 1.4 to 1.1. Conclusion: Based on the decrease of all the investigated parameters, it is possible to say that extrusion is currently an excellent processing feed technique in aquaculture with a good level of respect for the environment.

## 1. Introduction

In the last two decades, new strategies for improving feeding techniques for the main fish species have been developing in order to reach global sustainability, in particular in an environmentally responsible manner. At the start of the new millennium, the European Union adopted the Water Framework Directive (2000/60/EC), which introduced a transnational vision of the management of the freshwater environment able to protect, manage and improve the quality of water resources across the EU. In 2018, the European Environment Agency [1] showed the efforts carried out by countries to monitor and assess the general health status of water bodies and stressed the importance of also promoting the sustainable use of available water resources in inland anthropic activities of the primary sector, such as agriculture and aquaculture, performed in strict accordance with natural waters. Many papers [2,3,4,5] have focused on the environmental impact of trout farms when uneaten food, fish catabolites (metabolites, ammonia gill excretion and carbon dioxide) and chemical treatments are not controlled and are discharged into the natural receiving waters. In the case of aquaculture specialized in the growing phase of rainbow trout (*Oncorhynchus mykiss*), the rearing technique is based on the use of flow-through systems which consist of raceways or concrete tanks with water constantly flowing down basins and the removal of waste at the outlet by gravity and the water current [6].

In aquafeed, two elements have mostly contributed to improving aspects of change concerning fish feeding, in particular, in rainbow trout: The use of the feed extrusion technique and the adoption of restrictive environmental rules. First, regarding rainbow trout feeding, the best technique is based on the fact that the feed distribution must be carried out until reaching a level close to satiety [7]. In fact, the feed conversion index and the protein efficiency index improve when rainbow trout are fed with a rationing level equal to 70% of the “ad libitum technique” [7]. This technique drastically reduces the amount of food not eaten by trout.

Feed quality also directly affects water quality, because a proportion of the feed intake by fish is returned to the environment as metabolites or soluble by-products of metabolism [2]. In this context, two main levels of measures (legislative and productive) have been adopted. Restrictive environmental rules of EU countries have imposed settling areas to eliminate solid waste, whilst new feeding technologies have also been considered. The evaluation of the impact of rainbow trout farming on the receiving water quality needs to take into consideration the water quality monitoring over a long period, based on a decade [8] or two years, in different flow-through farms [4]. Concerning feeding technologies, nowadays, modern systems use extruded feed instead of pellet feed. In particular, the metabolite status of fish depends on the degree of gelatinization of the feeding starch. In the modern aquatic system, the removal of metabolites can be decreased using formulations with ingredients able to help bind metabolite matter, allowing these particles to be more easily and thoroughly removed from the water [9,10,11].

Furthermore, small feed portions not consumed are decreased by a smart diet formulation and processing; nowadays, extruded feed represents the best solution, instead of the pellet one, which was previously the common feeding strategy [9,10,11]. The digestibility of the extruded feed increases up to 96% relative to the raw wheat starch, while, in the case of pelleted feed, this raw material is digestible to approximately 54% [12]. Moreover, the catabolic residues emitted by rainbow trout, fed with extruded feed with gelatinized starch, are easy to remove from the water, thanks to their sedimentation [11]. Therefore, extruded feed also enables high environmental sustainability due to the increase of digestibility and stability in water, with a consequent reduction of suspended solids and nutrients. With regard to the rearing environment and its traits, it is appropriate to consider how it influences the conditions of well-being. Correct water exchange is essential: Water dilutes and removes the catabolites of fish, as well as feed residues, thus reducing the exposure of the farmed subjects to dangerous nitrogen compounds which are also able to negatively influence the state of well-being.

To check that the water quality is suitable for the farming of rainbow trout, a multitude of physical and chemical parameters must be considered [13]. In the case of suspended solids, their concentration is particularly important, because they directly influence the water turbidity, which can prevent the vision of the fish and finally compromise their life cycle. According to Boyd [3] and Becke [14], water turbidity values higher than 400 mg/L can cause thickening and deformation of the gill filaments, with trout consequently suffering. At the same time, dissolved organic substances, including the sedimentable (undigested portion of the ration and food residues) and non-sedimentable (product of endogenous metabolism) solids in suspension, must be taken into consideration.

For these reasons, checking the water quality is of primary importance for the welfare of trout. Inappropriate rearing conditions, such as inadequate space, excessive densities and poor feeding, can have strong negative repercussions for farmed fish species. Damaged, eroded or hemorrhagic fins are not only correlated with pathological events but also with inadequate environmental factors, connected to stress-related aspects such as a fish stocking density that is too high with a non-optimal water quality [15].

Considering the use of diets administered to salmonids, many papers have focused on the emission of catabolites in the external environment. According to European Environment Agency [16], 15–25% of the total food energy is lost in ammonia and urea through the gills and is released into the environment.

Most of the papers on this topic have focused their attention on relationships between feeding strategies, animal welfare and environmental sustainability and in the current literature, the correlation between water quality and productive performances of farmed rainbow trout is still a key point of discussion.

Based on these considerations, a study focusing on the trend of the most important water quality parameters in rainbow trout farming of central Apennine in terms of long-term activity (2009–2019) was carried out. Before the decade focused on in this study, the first historical monitoring of water quality took place in a trial carried out in 2004, when the owners started to re-think the farming technique adopted until then, based on high stocking densities (40 kg/m^3^), in order to produce a higher fish welfare status [17]. Considering this, the present study aimed to evaluate the suitability of changes adopted in rearing and feeding techniques to improve growth performance, sustainability and the water quality environment. The raceway water quality was monitored in terms of the Total Suspended Solids (TSS), Biochemical Oxygen Demand (BOD_5_), Chemical Oxygen Demand (COD), Total Ammonia Nitrogen (TAN), Nitrites (NO_2_-N), Nitrates (NO_3_-N), pH and Total Phosphorus (TP). These parameters were investigated over a decade, starting in 2009 and were compared to the respective annual values of the 10 years after, until the year 2019, in order to show differences in the water quality and feed conversion rate.

## 2. Materials and Methods

### 2.1. Trout Farm Description and Water Quality Sampling

A study on the water quality trend was performed on a rainbow trout farm, located in the Apennine area of central Italy, based on raceways in parallel (120 m^3^ each).

During the time of the study, the genetic line of rainbow trout did not change and it was directly controlled by the farm’s owners, who had a broodstock farm in a different area from which fishes for fattening came and were selected.

On the rainbow trout farm in which the study was conducted, the feeding technique was the same during the entire experimental period (2009–2019) and the feeding rate was the same for fish at the same life cycle stage. The feed was distributed with a semi-moving wagon up to the level close to satiety and at the same time during the day, twice a day.

The water supply system used on the farm allowed at least one complete daily water change. The inlet water came to the farm system with a constant velocity of 0.25 m/s and flowed through four parallel raceways.

A water quality sample was obtained monthly during the last decade (2009–2019) in correspondence with the lagoon basin (outlet water) below the raceways and receiving downstream. All samples were collected in the early morning before feeding using a polypropylene bottle with a screw cap [18].

In order to monitor the water quality, different physicochemical parameters were investigated. The dissolved oxygen, temperature and pH were measured using portable electronic devices (YSI mod. 55 and 60). At the same time, five samples of 500 cm^3^ water were collected for laboratory-based determination of the following parameters: total suspended solids (TSS), biochemical oxygen demand (BOD_5_), chemical oxygen demand (COD), total ammonia nitrogen (TAN), nitrites (NO_2_-N), nitrates (NO_3_-N) and total phosphorus (TP). Nitrogen (N) compounds and TP were determined using a spectrophotometer (Hach mod-2005, Hach Company, Loveland, USA), following the American Water Works Association and Water Pollution Control Federation of American Public Health Association (APHA) standard methods [19]. TSS was recorded following official methods [20]. BOD_5_ and COD were determined according to IRSA-CNR [20] and ISPRA [18] methods, respectively.

In order to quantify the results of the waste water of the rainbow trout farm, the quality of the inlet water was also assessed by analyzing the values of TSS, COD, BOD_5_, NO_2_-N, NO_3_-N and TAN, using the same laboratory methodology as described for the outlet water. Each sample of inlet water was collected using a sampler that conducted an average sampling process over three hours. During each year of the study, inlet water was collected seven times, in the months of January, April, May, June, August, October and December. The inlet water showed constant values for every parameter from 2009 to 2019, with the following averages: TSS, 5 mg/L; COD, 5 mg/L; BOD_5_, 5 mg/L; NO_2_-N, 0.09 mg/L; NO_3_-N, 1.4 mg/L; TAN, 1.5 mg/L. Since these values were always suitable for farming rainbow trout, additional investigations and more frequent water samples were not necessary, allowing us to focus our attention on the analysis of the outlet water.

### 2.2. Experimental Design and Rearing of the Rainbow Trout

In order to summarize the methodology that was followed, Figure 1 shows the experimental design step by step. With reference to the first year (2009) and the final year (2019) of the considered decade, fish growth and water quality assessments were also evaluated by considering the balance of nitrogen and phosphorus released in waters. In a fish plant, the productive cycle starts from young rainbow trout, with an average body weight of 90 ± 2 g, which are grown at a stocking density of 20 kg/m^3^ until reaching the market size (350 g). In particular, at the beginning of the life cycle, the fish density is low, about 8 kg/m^3^ of water, in order to limit the stress of young trout and prevent gill diseases or bacterial infections. This density then increases with the increasing size of the trout until the end of the cycle, when it reaches 20 kg/m^3^ of water.

Fish received extruded feed (closed formula) that was 4.5 mm in diameter and was manufactured by the same fish farmer who had vegetable feedstuffs available at his own land farm. The proximate composition of the two feeding types (pellet, extruded) employed to feed rainbow trout during the decade is reported in Table 1.

The final mean body weight (g) was recorded and the feed conversion ratio (FCR) was calculated from the amount of food consumed (kg) and the total biomass (kg) gained:

FCR = kg food consumed/kg final biomass − (kg initial biomass + kg sampled fish) + mortalities.

In order to evaluate the mass balance of nutrients released into the water, the amount of N and TP derived from the administered feed and the amount retained by rainbow trout were considered, as shown in Table 2, by comparing the budget of these compounds, expressed as the seasonal mean of the first year (2009) and the last year (2019) of study and applying the coefficients indicated by Bureau et al. [21] and applied by other authors [4].

### 2.3. Statistical Analysis

All collected data on the outlet water quality were analyzed to determine whether there were significant differences over the 10 considered years. For this aim, the months of the year were divided to define the four seasons: winter included the months of December, January, February and March; spring was represented by April, May and June; summer included July, August and September; and autumn included October and November.

After all seasonal data were collected, the mean of each parameter was calculated season by season for every year and data were finally organized as graphics.

The seasonal mean of the investigated parameters (TSS, BOD_5_, COD, NO_2_-N, NO_3_-N, TAN, TP and pH), recorded per year, was subjected to one-way analysis of variance (ANOVA) using the General Model Procedure of SPSS 25 (IBM Corp., New York, NY, USA) [22], in order to assess if data means were statistically different within the season of the different years. Significance was considered if *p* < 0.05 and the means were compared using the Student-Newman-Keuls (SNK) test.

## 3. Results

The results of the water quality parameters measured in the outlet lagoon basin in the 10 year-study are reported in Figure 2, Figure 3, Figure 4, Figure 5, Figure 6, Figure 7, Figure 8 and Figure 9, comparing the means of the same season of the 10 years.

The TSS (Figure 2) content showed a significant decrease when passing from the first seasons of sampling to the last years. In winter, TSS exhibited a significantly decreased mean value from 35.5 ± 5 mg/L in 2009 to 13 ± 3 mg/L in 2019, with an intermediate reduction notably shown in 2010–2012 (26.75–19.8 mg/L); in 2013 (14.5 ± 2 mg/L), TSS reached the lowest values until 2019. In spring, the same trend was observed, when the means decreased from 29 ± 3 and 27.3 ± 2.6 mg/L (2009–2010, respectively) to around 14.3 mg/L, which was maintained from 2014 to 2019. In the summer season, the recorded means were 37.20 ± 5 mg/L in 2009 and 35.4 ± 4 mg/L in 2010, without significant differences, whereas a marked reduction was observed from the year after (2011, 23.4 ± 3 mg/L), gradually diminishing from 2012 and remaining at around 16 mg/L until 2019. In Autumn, the highest TSS content was recorded in 2010 (35.5 ± 3.9 mg/L), being significantly different from the analyses of the first year of monitoring (2009, 26.5 ± 2 mg/L) but from 2011, the mean value was significantly lower and was maintained at around 16.00 ± mg/L until the end of the study. In particular, considering the seasonal average of TSS detected in Autumn, there was a reduction of almost 55% of this parameter.

Considering the trend of BOD_5_ (Figure 3), there was a marked improvement in 2019 compared to the 10 years before. In winter, BOD_5_ recorded a significant decrease, passing from the mean value of 9.5 ± 0.9 mg/L in 2009 to that of 7 ± 0.8 mg/L in 2010 and further decreased to around 3.7 ± 2 mg/L in the following years, until 2019. In spring, the highest levels were determined in 2009 (7.33 ± 1 mg/L) and 2010 (7.06 ± 0.7 mg/L); after this, the means significantly stayed at around 5.00 ± 0.7 mg/L until 2019. In summer, the highest levels of BOD_5_ were observed in 2009–2010 (8.7–9.3 mg/L), which were significantly different from all of the following means, being around 4.9 ± 1 mg/L until the end of the monitoring period. In autumn, a similar trend was also observed, with a notable difference between the first two years of sampling (7.18–6.72 mg/L in 2009–2010) and the other following years, being around 3.7 ± 0.7 mg/L until the end of the study.

In terms of COD determination (Figure 4), all four seasons showed a similar trend, resulting in a sharp reduction through the years. In winter, the highest levels were recorded in the first two years (2009, 32.5 ± 5.1 mg/L; 2010, 23.5 ± 4.4 mg/L), significantly decreasing in 2011 (13 ± 3.1 mg/L) and reaching the lowest mean in 2012, when the average of around 7.5 mg/L was maintained until 2019. In spring, the highest values occurred over the first two years (25 ± 1.3 mg/L) and then decreased in 2012–2013 (25–23.1 mg/L), until reaching the significantly lowest level in the last year (5 ± 0.2 mg/L). In summer, the first two years exhibited the highest levels (26.3–27 ± 1 mg/L). The mean value was decreased in 2011 (13.5 ± 0.3 mg/L) but was significantly higher than that in the following years, being around 7.5 ± 0.6 mg/L until the end of the study. In autumn, the mean of this parameter was 23.52 ± 0.5 mg/L in 2009 and 21.14 ± 0.7 in 2010; then, it decreased to 12.5 ± 0.3 mg/L in 2011 and reached 6.8–5.51 mg/L in the last years, without notable differences.

Regarding the TAN parameter (Figure 5), a clear reduction was observed in the range of the years taken into consideration. In the winter season, the data showed a significant decrease from the first year of study, from 1.15 ± 0.3 mg/L in 2009 to 0.1 ± 0.02 mg/L in 2019. In spring, TAN significantly decreased from 0.90 ± 0.2 mg/L in 2009 to 0.10 ± 0.02 mg/L in 2019. During summer time, a significant reduction was observed, that is, the value was 0.30 ± 0.01 mg/L in 2009 and significantly decreased from 2010–2012 (0.21–0.2 mg/L) and then further decreased to 0.1 ± 0.02 mg/L in 2019. In autumn, the TAN value decreased from 0.21–0.22 mg/L in 2009 and 2010 in the following years (2011–2019), when a mean value of 0.1 ± 0.02 mg/L was constantly recorded.

In terms of the NO_2_-N parameter (Figure 6), in winter, the mean values (0.1–0.09 mg/L), detected in 2009–2010, significantly decreased to 0.02 mg/L throughout the following years. In spring, an opposite trend was noted, with the lowest concentrations (0.01 mg/L) occurring in 2010–2011 in comparison to all other years (0.1 mg/L). In summer, a similar trend was observed, although the last year of monitoring showed the lowest mean (0.01 ± 0.002 mg/L) recorded among the years. On the contrary, in autumn, the TAN level detected in the two first years (0.1 mg/L) significantly decreased to 0.02 ± 0.001 mg/L in 2019.

In terms of the NO_3_-N parameter (Figure 7), a similar trend was observed in all the seasons, with a notable reduction emerging in 2013 and being maintained throughout the last years. In winter, the content dropped from 3.73–3.78 mg/L in 2009–2010 to only 1.22 ± 0.3 mg/L in 2019. In spring, the value significantly changed from 3.37 ± 0.5 mg/L in 2009 to 1.23 ± 0.3 mg/L in 2019. In summer, the value ranged from 3.90 ± 0.5 mg/L in 2009 to 1.50 ± 0.3 mg/L in 2019. In autumn, the concentration ranged from 3.60 ± 0.4 mg/L in 2009 to 1.30 ± 0.2 mg/L in 2019.

Considering the TP parameter (Figure 8), in winter, the values significantly decreased from 0.2 ± 0.01 mg/L in 2009 to 0.1 ± 0.002 mg/L in 2019. In spring, the highest values (0.19–0.2 mg/L) were recorded in the central part of the study (2014–2016); then, a reduction was observed, that is, 0.1 mg/L. In summer and winter, although there was a significantly higher level in 2016 (0.13 ± 00.2 mg/L), TP remained at a low value (0.1 mg/L) until 2019.

The pH parameter (Figure 9) did not show significant variation, remaining approximate neutral during the entire studied period and the obtained values were always in accordance with the range considered appropriate for rainbow trout [8,17].

At the end of the study, the FCR was 1.1 and significantly improved in comparison with the FCR obtained in 2009 (1.4). The survival rate was maintained at around 97% from 2012 onwards compared to the survival rate of the years before (94%).

In Table 2, the nutrient budget of outlet water is reported. In 2009, TAN was approximately 0.55 mg/L, whereas in 2019, the budget had an average value of 0.46 mg/L. The same trend was observed for the TP budget, which ranged from 0.011 mg/L (2009) to 0.009 mg/L over the 10 years.

## 4. Discussion

The aim of the current study was to investigate the impact of a new type of feed on water quality. Based on a comparison of the values of the analyzed parameters from 2009 to 2019, a notable decrease was observed in the TSS, BOD_5_, COD, TAN, NO_2_-N, NO_3_-N, pH and TP.

As other studies have shown [8,23,24], BOD_5_ is a very important parameter for assessing the water quality because it indicates the consumption of oxygen in the processes of indirect oxidation by the metabolic systems of aerobic microorganisms present in the water. Aside from the TSS level [24], BOD_5_ determination is essential for detecting the oxygen demand of the aerobic microbial flora required for the decomposition of organic substances present at a certain temperature in a defined time range. Therefore, it is an indicator that increases as the amount of organic substance to be mineralized increases.

The low values of BOD_5_ detected during 2019, together with the decreased COD levels, indicate that aerobic microorganisms prevailed over anaerobic ones and therefore, the self-purifying capacity of the watercourse can be considered good. The reduction of the COD parameter that exceeded 76.5% is proof that the technical and management interventions implemented at the breeding rainbow trout farm effectively resulted in an improvement in the environmental protection. This result is in agreement with data presented by Galezan [24], while Tahar [8] reported lower BOD_5_ values.

Based on analysis of the main water quality parameters of the trout farm over the last decade, a significant improvement in the rearing environment was shown. The significant reduction registered can be attributed to the different type of feed adopted, which changed from pelleted to extruded feed. Together with adequate quantities of oxygen dissolved in the water and through the nitrification process, a decreased amount of TAN can be converted into nitrites and nitrates. Nitrates are less toxic to fish over a long time period throughout the nitrification process [24]. A trend of the reduction of undesirable physicochemical traits was clearly observed when the new feeding type was introduced in 2011. Based on a comparison of the pellet feed with the extrusion technology, the digestible level of the diet was shown to have improved. This extruded manufacture drastically reduces the amount of food not eaten by trout and increases the feed efficiency [25].

The property of the extruded feed, which floats more than the pellet feed, increased its availability to the trout for a longer time in the water column before falling to the bottom of the raceways [9,26]. Fish can quickly intercept food without its dispersion and waste production, which occurs more frequently when using pellet feed because it sinks and degrades more easily and quickly in the water column.

In particular, concerning the two different feed techniques, that is, the pellet feed and the extruded feed, the processing style changed from the first to the second in 2011 but the raw materials remained the same. The fish meal and fish oil feedstuff were the same and both were of a high quality. The owner of the farm where the study was conducted monitored the feed quality through periodic laboratory analysis to ensure that the composition of the feed remained unchanged.

During the decade of the study, no differences in feed quality were shown and for the most important nutrients, no raw material changes were reported. For this reason, in the current study, the specific chemical analysis of the feed was not taken into consideration in order to stress the topic of water quality.

NO_2_-N represents an important stage in the oxidation of organic substances containing nitrogen. It is therefore a transitory form of nitrogen which is transformed into nitrates due to the bacterial activity in the presence of optimal quantities of dissolved oxygen in the water tanks.

Nitrates represent the final indicator of the degradation protein processes; the main sources are correlated with anthropic activity and in surface waters, the trend is usually seasonal.

Taking into consideration the TSS parameters detected in 2009 and comparing them with those of 2019, a significant reduction was clear, especially the trend observed during winter and Autumn.

According to a number of studies that have focused their attention on the relationship between feeding management and water quality [2,3,11,23,24], the significant improvement in water quality at this farm was due to the adoption of the modern type of feed based on the extrusion technique. In fact, compared to the pellet food, the new extruded feeding technique showed a greater stability in water and it allowed the food to be available to trout for a longer period of time. There are other positive elements to consider: The extruded feed had a higher fat absorption capacity and it was possible to know its specific weight. Therefore, it was easy to control the fish feeding and, as a consequence, the composition of the meat obtained from this fish. Our study supported the idea that extrusion is nowadays the best processing feed technique in aquaculture, as previously shown by other studies. Welker [26] compared the extrusion technique with the pellet feed by analyzing the water stability, fecal durability and digestibility and found the best results with the use of extruded feed. Similar results were reported by Tyapkova [9].

Moreover, in aquaculture, the extrusion technique positively affected the water quality: Food waste due to dust, breaks and “leaking/leaching” was decreased, improving the availability of nutrients for fish and minimizing the environmental impact. Another element of the extruded feed is that it is characterized by a low sedimentation rate. To ensure feed suitable for trout, a low sedimentation rate is one of the main physical properties to consider because it means the extruded feed is available to the fish for a long time; consequently, the fish can rapidly intercept the food without its dispersion and waste production. On the other hand, the use of pellet food means that feed is crumbled, which involves the fragmentation of food cylinders. As a consequence, this could lead to a loss of food (even if decreased). The feed was pulverized and this waste could remain at the bottom of the raceway water, becoming a possible source of water degradation, as well as microbial contamination.

Based on data concerning the monitoring of the rearing waters and considering the loads of N and TP of extruded feed, which took place in the last 10 years, it is possible to say that the improvement of the feed, administered to rainbow trout, is proof of the excellent quality of the ingredients used in fish diets [27,28].

Our data are in accordance with previous studies [3,4,5,8] that estimated ammonia and phosphorus emissions in water tanks from dietary analysis. In particular, we found lower TAN values than those reported by Moraes [5] and Aubin [4], which is a sign of the high feed quality used on the farm involved in our study. Concerning the *p* values, our results are in line with those reported by Moraes [5] and are lower than the values reported by Aubin [4] and Dalsgaard [23].

In this study, the amount of TAN and TP present in the tanks that in the past had hosted trout that received pelleted feed was compared to current tanks that host trout fed with extruded feed. The significantly decreased TAN and TP loads are proof of an improvement in diet quality, with a consequent benefit for the environment and the health of the trout.

This study provides insights into the connection between fish feed and water quality, that is, the fish habitat, which has to be suitable for producing healthy fish and as waste water, which returns to natural water bodies, taking into consideration specific parameters to investigate the water quality. In particular, in our study, an improvement trend was mostly demonstrated by the TSS, confirming that feed manufactured by the extrusion technique was of high quality. In particular, a notable positive change in the general trend was demonstrated after the new extruded food processing style was introduced, which occurred in 2011. In fact, all parameters showed an important decrease after the adoption of the extruded feed. More specifically, in the last years of the analyzed decade, a more favorable feed conversion rate was shown with respect to when pellet feed was administrated, resulting in better growing performances exhibited by trout whilst saving approximately 40% of the feed. In fact, the feed conversion index and the protein efficiency index improve when rainbow trout are fed with a rationing level equal to 70% of the “ad libitum technique” [7].

Another important change, which could have contributed to improving the efficiency feed conversion, could be justified by the fact that, on the farm, only trout larger than 90 g are reared, which do not require meal or crumbled feed. The fragmentation of crumbled diets leads to a more pulverized feed. In this case, the uneaten extruded feed, over 4 mm in size, is easier to remove than the pelleted diet.

Regarding the feedstuffs of the extruded feed, the inclusion of vegetable sources (soybean, wheat and pea) from the owners’ farm, located close to the fish plant, can be reported as an example of sustainability with positive effects on the production cost. On the fish farm, the fish stocking density was decreased (20 kg/m^3^) with respect to the first years of the decade, when rainbow trout were stocked at double the biomass. The choice to limit the culture density is the basis of the application of the “multisite” rearing technique, which aims to increase biosecurity, preventing diseases due to a vertical transmission of pathogens [29]. The decision of this plant to rear only rainbow trout starting from pre-fattened fish showed more advantages in terms of the survival rate, which was more satisfactory in the last years.

Concerning, more specifically, the fish farming density, before the decade focused on in this study and in 2004, the owners decreased the fish density in their raceways to improve the welfare of the rainbow trout. From 2009 to 2019, the fish density remained the same due to the effective improvement in animal welfare.

As previously shown by Welker [11], the correct management of tanks is essential to ensure good water conditions for trout and the appropriate use of water oxygenation systems contributes to maintaining optimal living conditions for these fish. This aspect could be connected to the improvement of the outlet basin receiving water from the raceways before being introduced into the stream.

Similar to what was observed by Becke [14], the pH was always close to neutrality and it was kept as stable as possible. This helped to minimize environmental stress and allowed nitrifying bacteria to effectively remove the nitrogen that accumulated in the sediments. Conversely, in other studies, the pH values were closer to 6 [5] or between 6.5 and 7.5 [8], which means that the effluent water was more acidic than the water considered here.

On the farm, the workers carried out normal system control every day, in order to ensure an oxygen level in the water that never dropped below 85% of the saturation level and its concentration was never less than 5 mg/L. The water supply system used on the farm allowed at least one complete daily water change, which directly affected the water chemical–physical parameters. Furthermore, the receiving water body has benefitted from these modified techniques, in particular the outlet basin, which was populated by floating macrophyte duckweed (*Lemna* spp.) that are known to be suitable for wastewater treatment and for food for pigs and poultry [30,31,32]. The duckweed filters nutrients, with a reduction of the eutrophic load of the water; it is an excellent purifying plant and absorbs the nitrogen in the water by eliminating nitrates [30,31].

Another aspect to consider is that, by comparing the parameters of the inlet water with those of the outlet water, which are the focus of the current study and due to the fact that the inlet water showed the same values over the 10 years of the study, it was possible to affirm that there was an improvement in the farming water quality. In fact, as shown before, the values of TSS, BOD_5_, COD, TAN, NO_2_-N and NO_3_-N were lower in 2019 than 10 years before.

Many papers have focused on the environmental impact of trout farms on the natural receiving waters in terms of uneaten food, fish metabolites and chemical treatments. For example, in a study carried out by Tahar et al., in 2018 [8], inlet and outlet concentrations of water parameters for four consecutive flow-through rainbow trout farms over a ten-year period were analyzed in Ireland to characterize the impact of each fish farm on the water quality as a function of their production and to identify any seasonal variability.

## 5. Conclusions

In the present trial, water parameters were investigated to determine whether the new feeding strategy based on extruded feed changed the water composition. The analyzed parameters (TSS, BOD_5_, COD, NO_2_-N, NO_3_-N, TAN and TP) showed an important improvement from 2009 to 2019; the pH parameter did not show important variation during the studied period.

Water is effectively the habitat where rainbow trout live so it must be monitored and preserved as best as possible in terms of temperature and physico-chemical parameters and quality. All these aspects are the basis of ensuring high-quality rainbow trout farming. Sustainable management, together with a genetic program of rainbow trout specimens employed, based on selected fish showing the best performance in feed efficiency, will be the next challenge to further improve the fish performance and environment.

Considering what has been discussed, the feedback of these results should be considered a significant index of improvement in the quality of the environment and the adoption of the modern formulation certainly contributed to obtaining this result.

## Figures and Tables

**Figure 1 animals-10-01541-f001:**
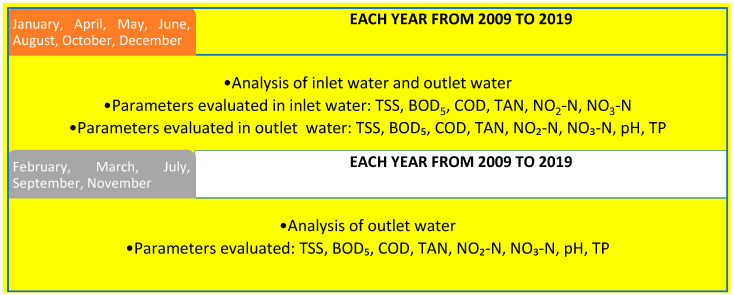
Diagram showing the experimental design adopted during the 10 years of the study.

**Figure 2 animals-10-01541-f002:**
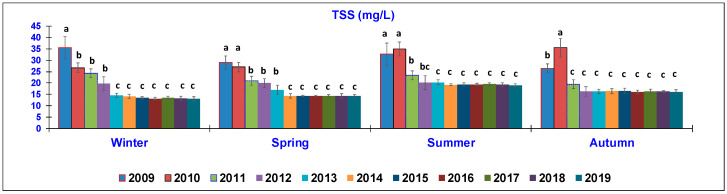
Trend of outlet water Total Suspended Solids (TSS) (mean ± standard deviation) seasonally determined during the 10 year-study. Different letters (a, b, c) per season show significant differences (*p* < 0.05) among the 10 years of sampling.

**Figure 3 animals-10-01541-f003:**
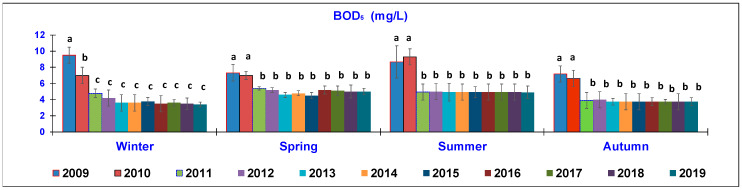
Trend of outlet water Biochemical Oxygen Demand (BOD_5_) (mg/L) (mean ± standard deviation) seasonally determined during the 10 year-study. Different letters (a, b, c) per season show significant differences (*p* < 0.05) among the 10 years of sampling.

**Figure 4 animals-10-01541-f004:**
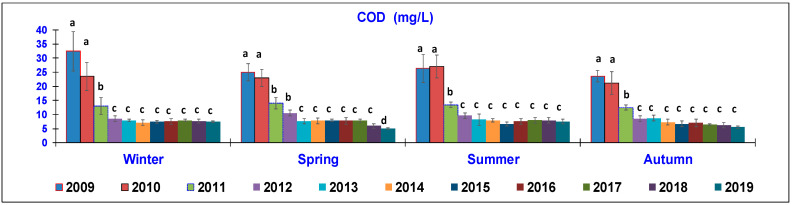
Trend of outlet water Chemical Oxygen Demand (COD) (mg/L) (mean ± standard deviation) seasonally determined during the 10 year-study. Different letters (a, b, c, d) per season show significant differences (*p* < 0.05) among the 10 years of sampling.

**Figure 5 animals-10-01541-f005:**
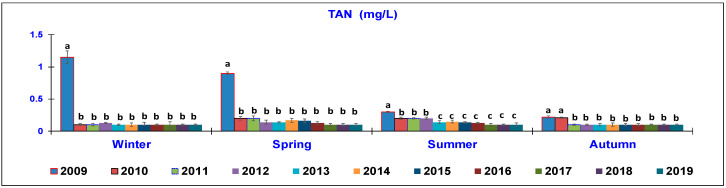
Trend of outlet water Total Ammonia Nitrogen (TAN) (mg/L) (mean ± standard deviation) seasonally determined during the 10 year-study. Different letters (a, b, c) per season show significant differences (*p* < 0.05) among the 10 years of sampling.

**Figure 6 animals-10-01541-f006:**
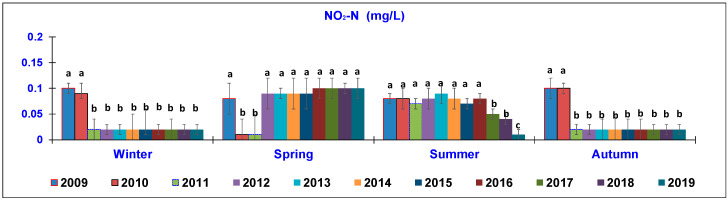
Trend of outlet water Nitrites (NO_2_-N) (mg/L) (mean ± standard deviation) seasonally determined during the 10 year-study. Different letters (a, b, c) per season show significant differences (*p* < 0.05) among the 10 years of sampling.

**Figure 7 animals-10-01541-f007:**
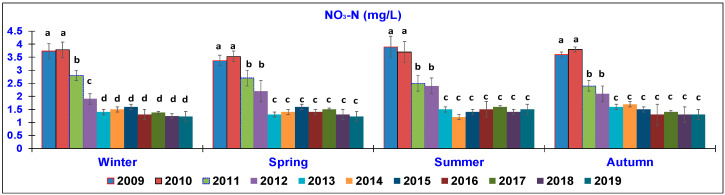
Trend of outlet water Nitrates (NO_3_-N) (mg/L) (mean ± standard deviation) seasonally determined during the 10 year-study. Different letters (a, b, c, d) per season show significant differences (*p* < 0.05) among the 10 years of sampling.

**Figure 8 animals-10-01541-f008:**
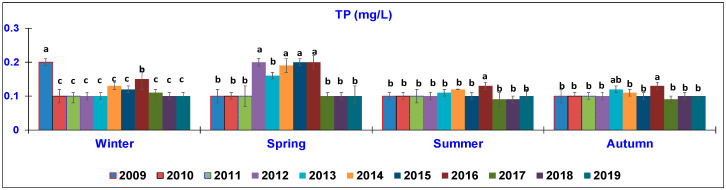
Trend of outlet water Total Phosphorus (TP) (mg/L) (mean ± standard deviation) seasonally determined during the 10 year-study. Different letters (a, b, c) per season show significant differences (*p* < 0.05) among the 10 years of sampling.

**Figure 9 animals-10-01541-f009:**
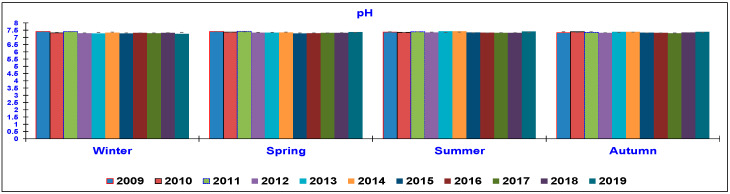
Trend of outlet water pH (mean ± standard deviation) seasonally determined during the 10 year-study.

**Table 1 animals-10-01541-t001:** Proximate composition of the feeds employed for the rainbow trout growing over the decade of water quality monitoring (2009–2019).

Feed	Pellet	Extruded
Chemical composition (%)		
Moisture	6.8	5.5
Crude protein	45.7	44.8
Crude lipid	16.0	21.0
Ash	6.7	8.4
Gross energy (MJ kg)	21.16	18.38

**Table 2 animals-10-01541-t002:** Total ammonia nitrogen (TAN) and phosphorus (TP) budget in the lagoon basin outlet water (mean ± standard deviation).

Year	TAN (mg/L)	TP (mg/L)
2009	0.55 ± 0.03	0.011 ± 0.001
2019	0.46 ± 0.02	0.009 ± 0.002

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
