# Peer review of "Improvement of the Water Quality in Rainbow Trout Farming by Means of the Feeding Type and Management over 10 Years (2009–2019)"

_animals, 2020, doi:10.3390/ani10091541_

Round 1
Reviewer 1 Report
This is an interesting study. The authors are to be commended for undertaking such a long-term study! Aside from the difficulties with the English grammar, word usage, and sentence structure, my major difficulties with this paper are with the methodology and conclusions. Specifically, there is not enough information (or control) to say that extruded feed was the reason for the changes in water chemistry. I need much more detail of the experimental design. Was the feeding technique constant? Was the feeding rate constant among the years? Was the number of times the fish were fed constant. Was the same rainbow trout strain (genetics) the same? Were loading densities the same? Was the extruded feed floating or sinking (and I assume the pellets were sinking)? What were the differences in feed ingredients? Any changes in any of these variables could have produced the results that are attributed just to extrusion technology. I think it is important to show, as the authors did, the improvement in rearing efficiencies (like FCR), but it is inaccurate given the methods as they are described to say that this was due solely to extrusion. And if the level of control was not present during the entirety of this study, it would still make for an excellent case study.
Author Response
Dear Reviewer,
I'd like to thank you for your comments and notes, that improve the article.
Point 1: Was the feeding technique constant? Was the feeding rate constant among the years? Was the number of times the fish were fed constant?
Response 1: Feeding technique, feeding rate and the number of times in which fish were fed were constant during all the experimental period. These details have been added in the Materials and Method section, from line 131 to line 137 on pg. 3.
Point 2: Was the same rainbow trout strain (genetics) the same?
Response 2: The genetic line stayed the same. More details have been added in the Materials and Method section, from line 131 to line 133 on pg. 3.
Point 3: Were loading densities the same?
Response 3: Loading densities were the same. At the beginning of the life cycle the fish density was low, about 8-10 kg /m3 of water, in order to limiting the stress of the young trout and avoiding gill diseases or bacterial infections. This density then grew up with the increasing size of trout until the end of the cycle, when it was 20 kg /m3 of water. These details have been added in the Materials and Method section, from line171 to line 174 on pg. 4.
Point 4: Was the extruded feed floating or sinking (and I assume the pellets were sinking)?
Response 4: The pellet feed was sinking, instead the extruded feed was floating. For this reason, as reported at lines 282-286 pg. 8, fish can intercept quickly the presence of food without its dispersion and waste production.
Point 5: What were the differences in feed ingredients?
Response 5: No differences in feed ingredients were conducted, only the form from pellet to extruded. See from line 287 to line 291 pg. 8.
Regards.
Reviewer 2 Report
Dear authors,
the manuscript entitled "IMPROVEMENT OF WATER QUALITY IN A 2 RAINBOW TROUT FARMING BY MEANS OF THE 3 FEEDING TYPE AND THE MANAGEMENT IN 10 4 YEARS (2009 - 2019)" determines the parameters of water quality for a time of 10 years where there was a change in the feeding of trout. Controlling these parameters and looking for ways to improve water quality in aquaculture is of utmost importance to avoid contamination of other bodies of water. This study presents important data, however, I recommend making improvements before it can be published.
You mentioned in the discussion (although it should also be written in methodology) that the feeding change was made in 2011, however if you observe the parameters of the water quality it began to improve dramatically since 2010. Therefore, The improvement in most parameters is exclusively due to the change in diet? or due to other factors not taken into account in this study?. Did you determine the quality of the water used in production (at the inlet)? This also influences the quality of the water in the effluent. There was no change in other factors for example the hydraulic retention time during the whole period ??? etc. A detailed description of the experiment is needed during this time period. The results presented in Fig. 1 are the average of how many samples? or did you take samples daily? please improve the methodology part. Statistics should be improved and presented in a better way. The discussion of results should be improved, based on a scientific soundness (why these results) and not only describing the results. Improve the conclusion. The conclusions must be supported by the results.
Specific comments:
line 7: check the affiliation numbers (you have only 1 and 3 and then you describe 1 and 2)
line 9: check font
line 33, 132: please use SI unit symbol (cm3, not cc) check all the document
line 52, 262, 286-287: check the reference format
line 126: which volume?
line 243-244: not true. Both ammonia and nitrites are very toxic
line 348: "were investigate" change to "were investigated"
Author Response
Dear Reviewer,
I'd like to thank you for your comments and notes, that improve the article.
Point 1: You mentioned in the discussion (although it should also be written in methodology) that the feeding change was made in 2011, however if you observe the parameters of the water quality it began to improve dramatically since 2010. Therefore, the improvement in most parameters is exclusively due to the change in diet? or due to other factors not taken into account in this study?
Response 1: The use of the pellet feed instead the extruded one was the only difference, so it should be the cause of the water quality improvement. In fact, other factors such as feeding technique, raw materials used in feed, fish density and the fish genetic line stayed the same, without any changes, during the period of the study. These aspects have been added in the article.
Point 2: Did you determine the quality of the water used in production (at the inlet)? This also influences the quality of the water in the effluent.
Response 2: The quality of the inlet water was assessed analysing the values of TSS, COD, BOD5, NO2-N, NO3-N and TAN, using the same laboratory methodology as descripted for the outlet water. During each year of the study inlet water was collected 7 times a year, and exactly every year in the month of January, April, May, June, August, October and December. This section has been added from line 156 to line 165 on pg. 4.
Point 3: There was no change in other factors for example the hydraulic retention time during the whole period ??? etc. A detailed description of the experiment is needed during this time period.
Response 3: Other factors did not change. More details have been added in the Material and Method section.
Point 4: The results presented in Fig. 1 are the average of how many samples? or did you take samples daily? please improve the methodology part.
Response 4: Water samples were collected monthly every year. Each time 5 samples of 500 cm3 of water were collected for laboratory analyses. This information has been added at line 147 pg. 4. The Materials and Methods section has been improved with more details concerning the methodology.
Specific comments:
Point 1 - line 7: Check the affiliation numbers (you have only 1 and 3 and then you describe 1 and 2).
Response 1: At line 7 pg. 1, the affiliation number are 1 and 2.
Point 2 – line 9: Check font.
Response 2: At line 9 pg. num. 1, the font has been completed.
Point 3 – line 33, 132: Please use SI unit symbol (cm3, not cc) check all the document.
Response 3: Review accepted. At lines 34 pg. 1 and 147 pg. 4 “cc” has been replaced with “cm3”.
Point 4 - line 52, 262, 286-287: Check the reference format.
Response 4 : Review accepted. At line 55 pg. 2, line 307 pg. 8, line 337 pg. 9 the references format has been checked.
Point 5 - line 126: Which volume?
Response 5: 5 samples of 500 cm3 water were collected. This information has been added in the text at line 147 pg. 4.
Point 6 - line 243-244: Not true. Both ammonia and nitrites are very toxic.
Response 6: Review accepted. The Reviewer is right, the concept was not exactly clear. I wanted to mean that nitrates are less toxic than TAN to fish in a long time thought the nitrification process. (Lines 275-277 pg. 8.)
Point 7 - line 348: "were investigate" change to "were investigated".
Response 7: Review accepted. The verbal form has been corrected at line 409 on pg. 10.
Regards.
Reviewer 3 Report
This is a worthy area of research as aquaculture strives to improve rearing practices and reduce environmental effects. However, it is already known that extruded feed had a number of improved characteristics over steam pelleted feed such as floating, more durable pellets, inactivation of antinutritional factors, improved nutrient digestibility and increased available energy. All of this has also been shown to affect water quality with the use of extrusion feeds shown to improve water quality. There are a few issues with this research and the design, first and foremost the researchers should have also taken and evaluated samples of incoming water to see if it changed seasonally or annually and what effect this might have on their downstream monitoring. Without this information the authors cannot decisively comment on the effects of density and diet. Also did the authors take samples of feed over time to determine what changes, beside going from steam pelleting to extrusion, were occurring in the feed, specifically formulation changes. Is the reported proximates in table 1 from 1 evaluation?
This paper is mainly descriptive but does not provide any information of water velocity? How many raceways the water flows through?, Water hardness and other parameters that are important.
Author Response
Dear Reviewer,
I'd like to thank you for your comments and notes, that improve the article.
Point 1: There are a few issues with this research and the design, first and foremost the researchers should have also taken and evaluated samples of incoming water to see if it changed seasonally or annually and what effect this might have on their downstream monitoring. Without this information the authors cannot decisively comment on the effects of density and diet.
Response 1: The quality of the inlet water was assessed analysing the values of TSS, COD, BOD5, NO2-N, NO3-N and TAN, using the same laboratory methodology as descripted for the waste water. Each sample of the inlet water were collected using a sampler that did an average sampling over 3 hours. During each year of the study inlet water was collected 7 times a year, and exactly every year in the month of January, April, May, June, August, October and December. This section has been added from line 156 to line 165 on pg. 4. Since these values were every time suitable to be used for a Rainbow trout farm, additional investigations and more frequent water samples were not necessary.
Point 2: Also did the authors take samples of feed over time to determine what changes, beside going from steam pelleting to extrusion, were occurring in the feed, specifically formulation changes.
Response 2: The owner of the farm where the study was conducted also monitor the feed quality through periodically laboratory analysis. During the decade of the current study no differences in feed quality were shown and in the most important nutrients no raw material changes were reported. More details have been added in the article from line 287 to line 295 on pg. 8. The specific chemical analysis of the feed was not taken into consideration in order to stress the topic of water quality.
Point 3: Is the reported proximates in table 1 from 1 evaluation?
Response 2: Every time the water samples were been withdrawn, 5 samples of 500 cm3 of water were collected for laboratory analyses. This information has been added at line 147 on pg. 4.
Point 4: This paper is mainly descriptive but does not provide any information of water velocity? How many raceways the water flows through?, Water hardness and other parameters that are important.
Response 4: The water velocity was 0.25 m/sec and it flowed through 4 parallel raceways. This information has been added in the article from line 138 to line 140 on pg. 3.
The inlet water was also investigated analysing the same values of the outlet one: TSS 5 mg/l; COD 5 mg/l; BOD5 5 mg/l; NO2-N 0,09 mg/l; NO3-N 1,4 mg/l; TAN 1,5 mg/l (from line 156 to line 165 pg. 4). The water hardness was also calculated, and it is 137 ± 6 mg /l;
this range was considered not to compromise the conducted investigation or animal welfare.
Regards.
Round 2
Reviewer 2 Report
Dear Authors,
the manuscript entitled "IMPROVEMENT OF WATER QUALITY IN A 2 RAINBOW TROUT FARMING BY MEANS OF THE 3 FEEDING TYPE AND THE MANAGEMENT IN 10 4 YEARS (2009 - 2019)" has been improved and most of my recommendations were incorporated or clarified. Please check the format of the units, sometimes you leave intermediate spaces and sometimes not and minor spell check is also recommended before publication. To enhance the manuscript quality, I would recommend a graph that exemplifies the experimental design which supports the methodology.
Author Response
Dear Reviewer,
I'd like to thank you for your suggestions, that really improve the article’s quality. Please see the attachment with my response.
Point 1: Please check the format of the units, sometimes you leave intermediate spaces and sometimes not and minor spell check is also recommended before publication.
Response 1: Review accepted. Format units and spell have been checked and modified along all the article.
Point 2: To enhance the manuscript quality, I would recommend a graph that exemplifies the experimental design which supports the methodology.
Response 2: Review accepted: A graph with the experimental design has been added as a PDF file. I’d like to thank you for this suggestion, that significantly improves the article’s quality.
Regards.